psychology/cognition

encoding, development, word learning, psycholinguistics, lexicon

**Author for correspondence:**
L. M. Henderson
e-mail: lisa-marie.henderson@york.ac.uk

# Growing up with interfering neighbours: the influence of time of learning and vocabulary knowledge on written word learning in children

S. Walker, M. G. Gaskell, V. C. P. Knowland, F. E. Fletcher, S. A. Cairney and L. M. Henderson

Department of Psychology, University of York, York YO10 5DD, UK

(iD) SW, 0000-0002-5613-5428

Evidence suggests that new vocabulary undergoes a period of strengthening and integration offline, particularly during sleep. Practical questions remain, however, including whether learning closer to bedtime can optimize consolidation, and whether such an effect varies with vocabulary ability. To examine this, children aged 8–12-years-old (*n* 59) were trained on written novel forms (e.g. BANARA) in either the morning (long delay) or the evening (short delay). Immediately after training and the next day, lexical competition (a marker of integration) was assessed via speeded semantic decisions to neighbouring existing words (e.g. BANANA); explicit memory was measured via recognition and recall tasks. There were no main effects indicating performance changes across sleep for any task, counter to studies of spoken word learning. However, a significant interaction was found, such that children with poorer vocabulary showed stronger lexical competition on the day after learning if there was a short delay between learning and sleep. Furthermore, while poorer vocabulary was associated with slower novel word recognition speed before and after sleep for the long delay group, this association was only present *before* sleep for the short delay group. Thus, weak vocabulary knowledge compromises novel word acquisition, and when there is a longer period of post-learning wake, this disadvantage remains after a consolidation opportunity. However, when sleep occurs soon after learning, consolidation processes can compensate for

weaker encoding and permit lexical integration. These data provide preliminary suggestion that children with poorer vocabulary may benefit from learning new words closer to bedtime.

## 1. Introduction

The acquisition of new words is vital for communication and language comprehension [1]. While some individuals accrue new vocabulary with ease, recognizing at least 10 000 words by the age of five [2], others face significant challenges, and this gap can persist or even widen with age [3]. This striking variability emphasizes the need for research to understand the factors that influence the process of word learning.

A complementary learning systems (CLS) account [4,5] posits that new words are initially acquired using the hippocampus to bind together cortical representations of form and meaning. In order to retain the word in long-term memory, it must be consolidated within neocortical systems, a process which has been shown to be influenced by sleep [6–8]. In support of this theory, newly learned spoken novel words often do not show hallmarks of an established lexical entry (e.g. engagement in lexical competition) until after a period of sleep [e.g. 9–11]. In these adult studies, lexical competition in spoken word recognition has been assessed via a pause detection task [9–11]. Participants were taught a novel sequence (e.g. 'daffodat') that overlapped with an existing word (e.g. 'daffodil'). They were then asked to make speeded decisions about the presence of a pause inserted near the offset of the existing word. If 'daffodat' has become *integrated* with existing vocabulary knowledge, it should shift the uniqueness point (the point in the phonemic configuration of a word at which it diverges from all other words) of 'daffodil' towards its offset. This should then interfere with the detection of a pause due to an increase in lexical activity at that point [9,11,12]. Interestingly, studies have found that lexical competition effects tend to strengthen after a period of sleep relative to an equivalent time spent awake in adults. Furthermore, overnight enhancements of lexical competition are associated with key sleep parameters known to support declarative memory consolidation, including sleep spindles and slow oscillations in adults [13]. Despite the weight of evidence pointing to overnight changes in lexical integration following spoken word learning, it is also important to acknowledge evidence that supports lexical integration in the absence of sleep [14], particularly from studies using eye-tracking to assess activation of competitor words (see [15], for a review).

A similar time course of lexical integration has been observed in adult studies of orthographic word learning. For instance, Bowers *et al.* [16] manipulated the presence of orthographic lexical neighbours for a series of hermit base words (i.e. words with no single-letter substitution, deletion, transposition or addition neighbours), to examine the time course by which novel words take on the characteristics of existing words. Bowers *et al.* taught participants new written pseudowords that substituted one internal letter of the hermit base word (e.g. BANARA from BANANA). This should have the effect of changing the orthographic neighbourhood size of the base word from zero (a hermit) to one (a non-hermit), and when the new word has been lexically integrated, this should work to slow down responses to the base words. To explore such competition effects, Bowers *et al.* used a semantic categorization task where participants were asked to identify whether the base word belonged to a natural or artefact category. On the day after training (but not before), response times (RTs) on this task were significantly slower to the non-hermit words whose neighbourhood count had been increased to one, in comparison to hermit words whose neighbourhood count remained at zero. This again implicates the role of offline consolidation in the integration of novel words into the mental lexicon. Wang *et al.* [17] adapted Bowers *et al.*'s methods to demonstrate that lexical competition effects emerged 12 h after training if participants were trained in the evening and tested the next morning after sleep, but not if participants learned the items in the morning and were tested in the evening after a day awake. Studies with children have also suggested that sleep plays a pivotal and active role in the consolidation of new spoken words earlier in development (e.g. [18–21]). Indeed, there is some evidence of *greater* overnight changes in lexical competition in response to spoken word learning in children than in adults [22]. These studies have also shown substantial improvements in cued recall performance for newly learned words over sleep, above and beyond the effects of repeat testing [18,23].

However, there is a relative lack of evidence for the role of sleep in written word learning in development. In one exception, Tamura *et al.* [24] trained children aged 9–11 years on orthographic novel competitors via storybook exposure. Regardless of whether direct instruction was incorporated alongside incidental exposure, 'lexical engagement' emerged only after sleep. Tamura *et al.* captured

lexical engagement via the emergence of a prime-lexicality effect. That is, the finding that form-related non-word primes can facilitate lexical decision responses to word targets (e.g. ANPLE–APPLE), but that form-related word primes can cause inhibition (e.g. AMPLE–APPLE) (see [25]). The logic here is that once a novel word has become lexically integrated, there should be a reduction in facilitation, presumably as a consequence of competition emerging for the non-word prime. This measure of lexical competition differs from that captured by pause detection and semantic categorization tasks, not least because the learned non-words are presented as primes during the task. Such cueing could facilitate parallel activation of both representations (e.g. ANPLE and APPLE) (see [22], for discussion), whereas pause detection and semantic categorization only require responses to the existing words and so, arguably, capture more automatic lexical competition. No studies, to our knowledge, have used semantic categorization as a marker of lexical competition following orthographic word learning in children. Thus, while there is general evidence of developmental stability in the importance of consolidation processes for word learning, there is less clarity on the time course by which *automatic* lexical integration effects emerge following orthographic word learning.

There is also striking heterogeneity in the rate of vocabulary growth in developing populations that is not well explained by current theoretical models. One potential source of individual differences concerns the protracted development of the hippocampus over the school years [3,26]. In their review, James *et al.* [3] proposed that the maturing hippocampus may place constraints on vocabulary acquisition during childhood, supported by a correlation between hippocampal volume and language ability that strengthens with age [27]. James *et al.* posited that the more immature the hippocampus, the more likely it will struggle to retain information, and subsequently be more prone to the effects of interference. Therefore, if sleep aids the process of consolidation, as indicated by the CLS account, it may be that during development, sleeping soon after learning is particularly beneficial.

Investigations of the optimal time for learning prior to sleep have mainly focused on adults, and evidence is mixed. It has been claimed that sleeping shortly after encoding is beneficial [28–32]. For instance, McGregor *et al.* [30] found that memory for new word meanings improved over a week but only for young adults who were trained in the pm, and not in the am. Such findings are consistent with the view that sleeping soon after learning circumvents interference and forgetting of newly encoded memories over periods of wake [28]. Conversely, others report that a period of wake prior to sleep can be useful [33,34], perhaps allowing for new memory traces to be strengthened via wake-based processing prior to consolidation. In line with this, Walker *et al.* [35] found that longer gaps between learning and going to sleep were associated with *better* explicit memory for novel words, but only one week after training. It was argued here that wake-based processing prior to sleep (in combination with repeated testing) may aid longer-term retention in adults. However, there was no evidence that time between learning and sleep influenced lexical integration of new orthographic forms (as measured by a semantic categorization task, following Bowers *et al.* [16]).

Research on the role of post-learning wake time in children is lacking. For example, Hupbach *et al.* [36] reported that infants who did not nap soon after exposure to new grammatical regularities showed no evidence of remembering the new grammatical patterns when tested 24 h later (in comparison to infants who napped soon after learning). Sleep–wake consolidation studies that train children on new stimuli in the am or pm and test immediately after training and 12 h later do not lend strong support for a benefit of sleeping soon after learning (i.e. no overall pm condition benefits are reported by the authors in [18,37,38]); however, such studies cannot provide clear evidence on this issue, given that additional exposure close to sleep is provided at the 12 h test for the am condition. Thus, further research is clearly required to examine whether sleeping soon after learning is beneficial (or indeed detrimental) in school-aged children.

Relevant to the influence of post-learning wake time on consolidation of new vocabulary is the extant vocabulary knowledge that a child brings to the task. Previous research has shown that a child's oral language ability is strongly associated with word learning in a variety of contexts [3,39,40], with recent claims that novel words can benefit from local connections with existing phonological neighbours during word learning [41]. In a sample of children aged 7–10 years, Henderson *et al.* ([42]; see also [39]) reported positive correlations between expressive vocabulary ability and overnight changes in lexical competition between new and existing competitors, implying that children with superior vocabulary show greater consolidation benefits in terms of overnight lexical integration (although see James *et al.* [41] for evidence that such correlations may not be found in the case of written words). Thus, it appears that existing vocabulary knowledge can be linked to both the encoding and consolidation of new vocabulary. Speculatively therefore, any effect of time of learning prior to sleep may interact with vocabulary knowledge. More specifically, weaker vocabulary

knowledge may give rise to weaker encoding, leaving new memory traces at greater risk of wake-based interference and/or reducing the possibility of effective wake-based rehearsal prior to sleep.

## 1.1. The present study

The current study examined the effects of the delay between learning and sleep on the explicit learning and lexical integration of written novel words in children aged 8–12 years. Children of this age show variation in hippocampal maturity while simultaneously showing a peak in slow-wave sleep (SWS) duration [43] and enhanced consolidation processes (proposed to compensate for the constraints of the developing hippocampus) in vocabulary learning relative to adults [41]). Thus, this age range presents optimal conditions to explore variation in susceptibility to the delay between learning and sleep as well as sleep-dependent memory consolidation in development. Lexical integration was indexed by the emergence of lexical competition between novel and existing competitors (measured via a semantic categorization task, following [16,35]); explicit aspects of novel word memory were measured via recognition and cued recall tasks. Global vocabulary ability was measured via a standardized test of receptive vocabulary, to ascertain whether any effects of the time between learning and sleep interact with vocabulary knowledge.

The study had four aims. First, we aimed to replicate previous adult findings showing consolidation benefits of orthographic word learning in children [16,17,35]. We hypothesized that children would show overnight increases in lexical competition between novel and existing competitors, and overnight improvements in their ability to recognize and recall the novel words. Second, we examined whether the time between learning and sleep influences consolidation, with the prediction that learning closer to sleep would give rise to greater improvements in overnight consolidation in childhood [28,29,31,32,36]. Third, we predicted that overnight increases in lexical competition would be associated with better vocabulary knowledge based on claims that existing knowledge facilitates lexical integration [39,42]. A final more exploratory aim was to examine a potential interaction between vocabulary knowledge and the effect of time between learning and sleep on consolidation. Although uncertain as to the direction of this effect, we predicted that children with lower vocabulary might be more prone to the effects of wake-based interference (hence leading to greater benefits in learning closer to bedtime).

# 2. Material and methods

## 2.1. Participants

Sixty monolingual English-speaking children aged 8–12 with no known auditory, language or sleep impairment and/or psychological disorder were recruited via University and local adverts and local schools. All participants were paid a £10 Amazon voucher for their participation. One participant was unavailable on the day of the experiment, leaving $N = 59$. The experiment was approved by University of York's Department of Psychology Research Ethics Committee.

## 2.2. Materials

### 2.2.1. Stimuli

Twenty-eight hermit base words that had no substitution, transposition, deletion or addition neighbours were selected (based on [16]). Each word was a noun, and was either six or seven letters in length, with a CELEX frequency of 2–31 per million [44]. Fourteen words were taken from those used by Bowers *et al*. The further 14 words were selected via N-Watch [45] using the above criteria. For the purpose of the semantic categorization task (described below), half of the base words referred to naturally occurring items (e.g. BANANA) and half to man-made items (e.g. ANCHOR). Novel 'words' were constructed by substituting one internal letter of each base word to form a pronounceable pseudoword (e.g. BANARA) (see [16]). Any double letters were preserved in the novel words (e.g. SLEEVE to SPEEVE), and only the phoneme of the substituted letter was affected. All novel words had one neighbour only (the hermit base word).

Each child was taught one of two matched and counterbalanced lists of 14 novel words. This ensured that each base word gained a neighbour for half the participants and remained a hermit word for the other

half. Lists were matched pairwise for initial letter and no list contained more than one word with the same first letter. In each list, seven novel words were neighbours from the natural category and seven from the man-made category. An additional set of 28 six- and seven-letter words were created to be used as fillers in the semantic categorization task. The fillers were a mixture of hermit and non-hermit concrete nouns, with a CELEX frequency ranging between 0.4 and 82.6 counts per million.

### 2.2.2. Language measures

The British Picture Vocabulary Scale 3rd Edition (BPVS-3: [46]) was used to assess each child's level of receptive vocabulary. In this test, children are required to select one of the four pictures that matches a word spoken by the experimenter. Thus, the test provides a measure of vocabulary 'breadth' (i.e. an index of the quantity of words a child knows). The standard score of this measure for each child was used to create the continuous variable, *Vocabulary*. Importantly, there was a normal distribution of vocabulary scores, and no differences in vocabulary knowledge between the groups ($t_{51} = -0.44$, $p = 0.66$); groups defined below.

### 2.2.3. Sleep measures

Each child was asked to wear an actiwatch [47] on the night of Day 1 in order to check for any group differences in total sleep time (TST). Actigraphy data were analysed using ACT MILLENNIUM v. 3.60.0.1 [47]. The first epoch of 10 consecutive minutes of immobility after bedtime was used to identify time of sleep onset and the last epoch of 10 consecutive minutes of immobility before get-up time was used to identify time of sleep offset. TST was calculated as the number of minutes between sleep onset and sleep offset. No differences in TST were seen between the groups; groups defined below (long delay group: average TST = 493 mins; short delay group: average TST = 492 mins; $t_{51} = 0.05$, $p = 0.96$).

## 2.3. Procedure and design

A mixed design was used (figure 1) with the duration between learning and sleep (*Delay*; short versus long) as a between-participants factor and day of testing as a within-participants factor (*Day*: Day 1 versus Day 2). Participants were randomly allocated to either the short or long delay group. For the semantic categorization task, there was an additional independent variable, Word type (hermit versus non-hermit). The hermit words acted as a control, as the neighbouring pseudoword was not learned. For the non-hermit words, the neighbouring pseudoword had been learned, potentially changing the competition environment of the existing word (base word).

Training and testing on Day 1 took place at school for the long delay group (between approx. 8 am and 10 am), and at home for the short delay group (between approx. 4 pm and 6 pm). Testing on Day 2 took place at school between approximately 8 am and 10 am for both groups.[1] On Day 1, participants were asked to complete a Training phase. Immediately following this, they completed the Testing phase, which consisted of a vigilance task (to measure attention), semantic categorization task (to measure lexical integration), and a cued recall and a speeded recognition task (to measure explicit memory). On Day 2, children again completed the vigilance, semantic categorization, cued recall and speeded recognition tasks. During the night of Day 1, children were asked to wear an actiwatch to measure TST.

### 2.3.1. Training

Each child was exposed to each novel word in their allocated list 12 times via E-Prime 2.0 [48]. The 12 presentations consisted of (i) two passive trials, which involved passively reading each pseudoword which was displayed on the screen for 4000 ms before automatically moving on, (ii) five typing trials, where they were asked to type out each pseudoword as it appeared on the screen; as they typed the word changed colour, (iii) three letter-monitoring trials, where they were asked to press the spacebar if the letter 'T' was present in the pseudoword, and (iv) two trials where they were asked to press the spacebar when the pseudoword changed colour. All novel words were presented in

[1]Due to practical constraints, one child in the long delay group was tested at home on both days. In the short delay group, three children on Day 1 were tested in school, and one child on Day 2 was tested at home.

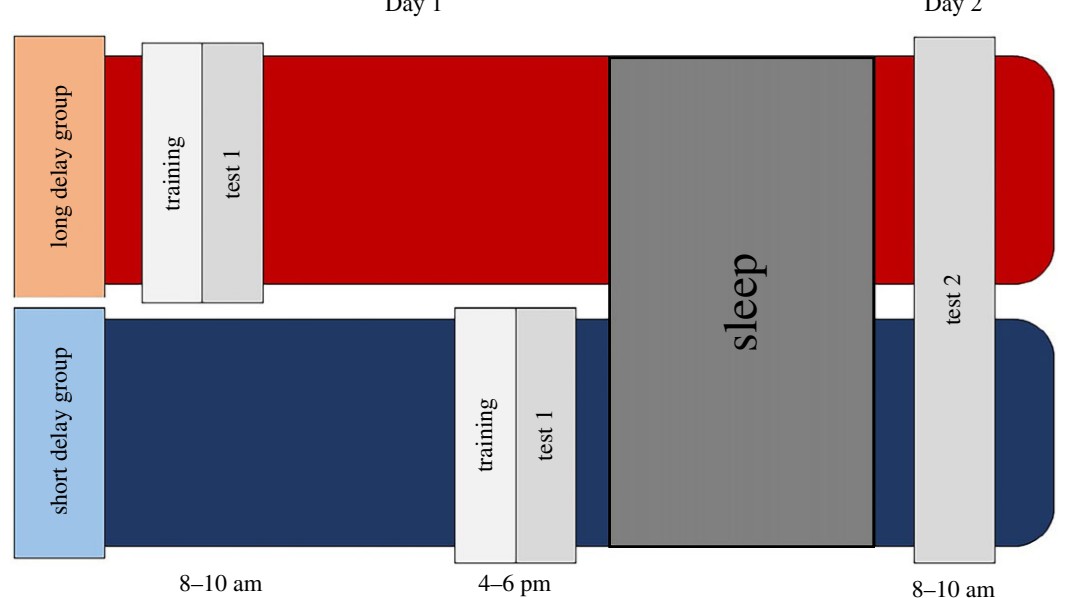

**Figure 1.** Experimental procedure.

upper case Calibri font. For each child, the different types of exposure to the list were presented in a fixed order. The novel words within each list were presented in a randomized order. For all trials, apart from the passive ones, no timeouts were set.

### 2.3.2. Testing

Children completed a vigilance task, a semantic categorization task, a cued recall task and a speeded recognition task during testing on both days. An additional check of children's understanding of the stimuli was completed in the final session. All tasks were conducted using E-Prime 2.0.

*Vigilance.* Children completed a psychomotor vigilance test (PVT; based on [49]). Children were presented with a star on the screen at random interstimulus intervals (1000–4000 ms, mean 2468 ms) and asked to respond as quickly as possible by pressing the spacebar. There were 90 trials in total. No timeout was set for this task. There were no group differences in the mean RT on Day 1 ($t_{51} = 0.15$, $p = 0.88$) or Day 2 ($t_{51} = 1.42$, $p = 0.16$), suggesting that attentional/circadian effects of time of day could not account for any findings.

*Semantic categorization.* Children were presented with all 28 base words and asked to respond 'yes' or 'no' to the question 'is this made by a human?' by pressing the Z (for yes) or M (for no) keys on a computer keyboard. Children were reminded on each trial which key represented yes and no. Each trial began with a fixation cross which was displayed for 500 ms, followed by the target word (e.g. BANANA) which was displayed until the participant responded. The words were presented in upper case, Calibri font. Feedback was provided during a short practice trial; but not during the experimental trials. There were a total of 10 practice trials and 56 experimental trials (14 of which contained the base words they had been exposed to during the Training phase). The 14 unlearnt base words acted as controls and the 28 filler words acted as distractors.

*Cued recall.* As all novel words were created by substituting one internal letter, for the cued recall task, the cue always displayed the first and last letter of the learnt novel word (i.e. a non-internal letter; e.g. B_ _ _ _ A). A list of 14 cues (created from each of the learnt novel words) was presented on a piece of A4 paper. Children were given the following instructions verbally by the experimenter:

> In this next task each of the words you learnt during training with us are listed on this sheet. However the middle letters are missing. We want you to try and remember as many of the new words as you can, and then fill in the missing letters. The number of letters missing is the same as the numbers of little lines. For each one we have given you the first and last letter. You have a maximum time limit of 5 minutes to complete this task.

*Speeded recognition.* In the speeded recognition task, participants were presented with all 28 novel words. Children were asked 'did you learn this word with us?' by using the Z (yes) and M (no) keys on the keyboard. Again, children were reminded on each trial which key represented yes and no.

There were 28 trials (14 novel learnt words and 14 novel unlearnt words). Therefore, all learned novel words were presented once. The trial structure (fixation cross, trial times, font type and size) was the same as that used in the semantic categorization task.

*Vocabulary check.* At the end of testing on Day 2, each child completed a four-alternative forced choice (4AFC) vocabulary check to assess their knowledge of each of the original base words (e.g. a banana). On each trial, four photo images were presented on a computer screen numbered 1–4 from left to right. Participants were asked to use the row of numeric keys 1, 2, 3 or 4 on a keyboard to select which of four pictures best represented the base word. The four photo images included an image of the base word referent (e.g. a banana), an image of an object semantically related to the base word (e.g. an orange), an image of an object phonetically similar to the base word (e.g. a bulldozer: the base word referent shared the same first phoneme and contained the same number syllables as the phonetically similar referent) and an unrelated object (e.g. a train). Accuracy was overall high on this task ($M = $ 97.8%, s.d. = 3.7%).

## 2.4. Analysis

### 2.4.1. Outlier removal

Prior to any analyses, outliers were identified using the following procedures: accuracy rates were averaged across sessions for the semantic categorization task. In accordance with our previous study in adults [35], participants with accuracy z-scores less than −2.5 were excluded ($n = 2$). For the speeded recognition task, a *d*-prime calculation based on hits (correctly identifying a learnt word) and correct rejections (correctly identifying an unlearnt word) was used to determine sensitivity. Outliers were detected using histograms. Six participants were identified and subsequently excluded from the analysis. Following this, for both the speeded recognition and semantic categorization tasks, a trim of 5000 ms was applied to all RT trials to exclude lapses. Any further accuracy z-scores less than −2.5 were excluded. Two participants were identified in the speeded recognition task and one participant was identified in the semantic categorization task.

For both the semantic categorization and speeded recognition tasks, within-subject RT outliers were classed as any trials 2.5 s.d. above a participants' mean reaction time. RTs less than 200 ms were classed as false responses and removed. For individual items, accuracy rates across all days were averaged separately for the semantic categorization and speeded recognition tasks. All item accuracy z-scores were less than 2.5.

Due to verbal task demands, one participant was removed from all analyses due to a BPVS standard score of 78 which falls below the 10th percentile. Three participants were removed as they did not complete the BPVS due to time constraints. The mean BPVS score across the remaining sample was 109.44 (s.d. = 10.35). Participants with BPVS z-scores greater than 2.5 were excluded (one participant with a BPVS score of 137 and one with a BPVS score of 84).

Once outlier removal was complete, the remaining data were analysed using R, with models fitted using the package lme4 [50] and figures made using ggplot2 [51]. Mixed-effects logistic regression models were used to model binary outcomes (cued recall accuracy) and linear mixed-effects models for continuous outcome data (semantic categorization and speeded recognition reaction times). For each dependent variable, fixed effects of *Delay* (short versus long; +0.5, −0.5), *Word type* (hermit versus non-hermit; −0.5, +0.5)[2] and *Day* (Day 1 versus Day 2; −0.5, +0.5) were included. Vocabulary scores were centred and scaled prior to model fit. Based on tests of normality, a log transformation was used to normalize the distribution of RTs [52].

The fixed-effects structure was simplified using a backwards selection procedure from the maximal fixed-effects structure, using likelihood ratio tests (LRT) and a liberal criterion of $p < 0.2$ to justify inclusion. Random intercepts and slopes were also justified using a liberal criterion for model improvement of $p < 0.2$ via LRT and added until no further model improvement could be established (i.e. $p < 0.2$) [53]. The *p*-values were provided by lmerTest [54].

Before establishing the best fitting fixed-effects structure, dfbetas were calculated to identify any strongly influential cases via the influence.ME package [55]. According to Nieuwenhuis *et al.*, [55], influential case analysis is necessary in addition to basic outlier removal, as it allows us to specifically examine the amount of influence a case exerts on the regression slope. Dfbetas were standardized and any participants with z-scores greater than ±3.29 were removed from that dataset; three from the

---

[2]Word type was only included in the model for the semantic categorization task.

**Table 1.** Mean (and s.d.) RT and lexical competition effects (non-hermit RT–hermit RT) for the semantic categorization task, for Day 1 and Day 2.

| delay | day | N | hermit (ms) | non-hermit (ms) | lexical competition (ms) |
|---|---|---|---|---|---|
| short | 1 | 24 | 1292 (342) | 1274 (275) | −18 (192) |
| | 2 | 24 | 1052 (225) | 1090 (250) | 38 (182) |
| long | 1 | 24 | 1243 (317) | 1311 (387) | 68 (181) |
| | 2 | 24 | 1027 (222) | 1051 (242) | 24 (141) |

semantic categorization task, one from the speeded recognition task and none from the cued recall task. We adopted a 3.29 threshold for the removal of influential cases (as opposed to the standard practice of a 2.5 threshold for outlier removal, as described below) to take a conservative approach and avoid the risk of unnecessary data loss.

# 3. Results

## 3.1. Lexical integration

The semantic categorization task was used to assess lexical integration. Descriptive statistics for all main study variables can be found in table 1. Accuracy was high across all conditions on Day 1 (short delay: $M = 85.7\%$ s.d. = 7.5; long delay: $M = 91.0\%$, s.d. = 5.9) and Day 2 (short delay: $M = 88.7\%$, s.d. = 7.4; long delay: $M = 91.0\%$, s.d. = 6.1).

RTs were faster overall on Day 2 ($M = 1055$ ms, s.d. = 179) than Day 1 ($M = 1280$ ms, s.d. = 192; $b = -0.19$, s.e. = 0.02, $p < 0.001$). Inspection of the competition effects in table 1 is suggestive of a pattern of increasing lexical competition overnight for the short delay group (+56 ms) and decreasing lexical competition for the long delay group (−44 ms). However, the Word type × Day interaction was not significant ($b = 0.01$, s.e. = 0.03, $p = 0.61$), meaning that, counter to our first prediction, there was no general overnight change in the competition effect. Moreover, our second and third predictions of three-way interactions between Word type × Day × Vocabulary ($b = -0.004$, s.e. = 0.03, $p = 0.90$) and Word type × Day × Delay ($b = 0.07$, s.e. = 0.05, $p = 0.19$) were also non-significant. However, our fourth (more exploratory) prediction of a four-way Vocabulary × Delay × Day × Word type interaction was significant ($b = -0.12$, s.e. = 0.06, $p = 0.041$). This interaction suggested that emergent lexical competition effects might depend on vocabulary and the delay between learning and bedtime, and was explored further using the R package *emmeans* [56] to examine the effect of vocabulary in each cell of the design (figure 2). For the long delay group, there was descriptively a general trend of faster responses for participants with higher vocabulary scores, but no differences between the slopes for hermits and non-hermits on either day (Day 1: $b = 0.02$, s.e. = 0.04, $p = 0.54$; Day 2: $b = -0.03$, s.e. = 0.04, $p = 0.35$). For the short delay group, the slopes for the different word types were more variable, with a significant difference emerging on Day 2 but not Day 1 (Day 1: $b = -0.02$, s.e. = 0.02, $p = 0.32$; Day 2: b = 0.04, s.e. = 0.02, $p = 0.033$). Although the strength of the statistical evidence here is slight (and it should be noted that the four-way interaction would not survive correction for multiple comparisons), the interaction provides tentative evidence that lexical competition effects were emerging on Day 2 for the short delay group, particularly for children with lower vocabulary.[3]

## 3.2. Explicit memory

The speeded recognition and cued recall tasks were used to assess explicit memory. Descriptive statistics for all main study variables can be found in table 2.

---

[3]Based on a reviewer's suggestion, we also interrogated the four-way interaction by running analyses on each of the delay groups separately. There was a significant Word type × Day × Vocabulary interaction in the short delay group alone ($b = -0.07$, s.e. = 0.03, $p = 0.025$). To follow this up, we divided the Vocabulary variable using a median split. The Word type × Day interaction was approaching significance in the Low vocab ($b = -0.29$, s.e. = 0.15, $p = 0.07$), but not the High vocab group ($b = 0.10$, s.e. = 0.13, $p = 0.44$).

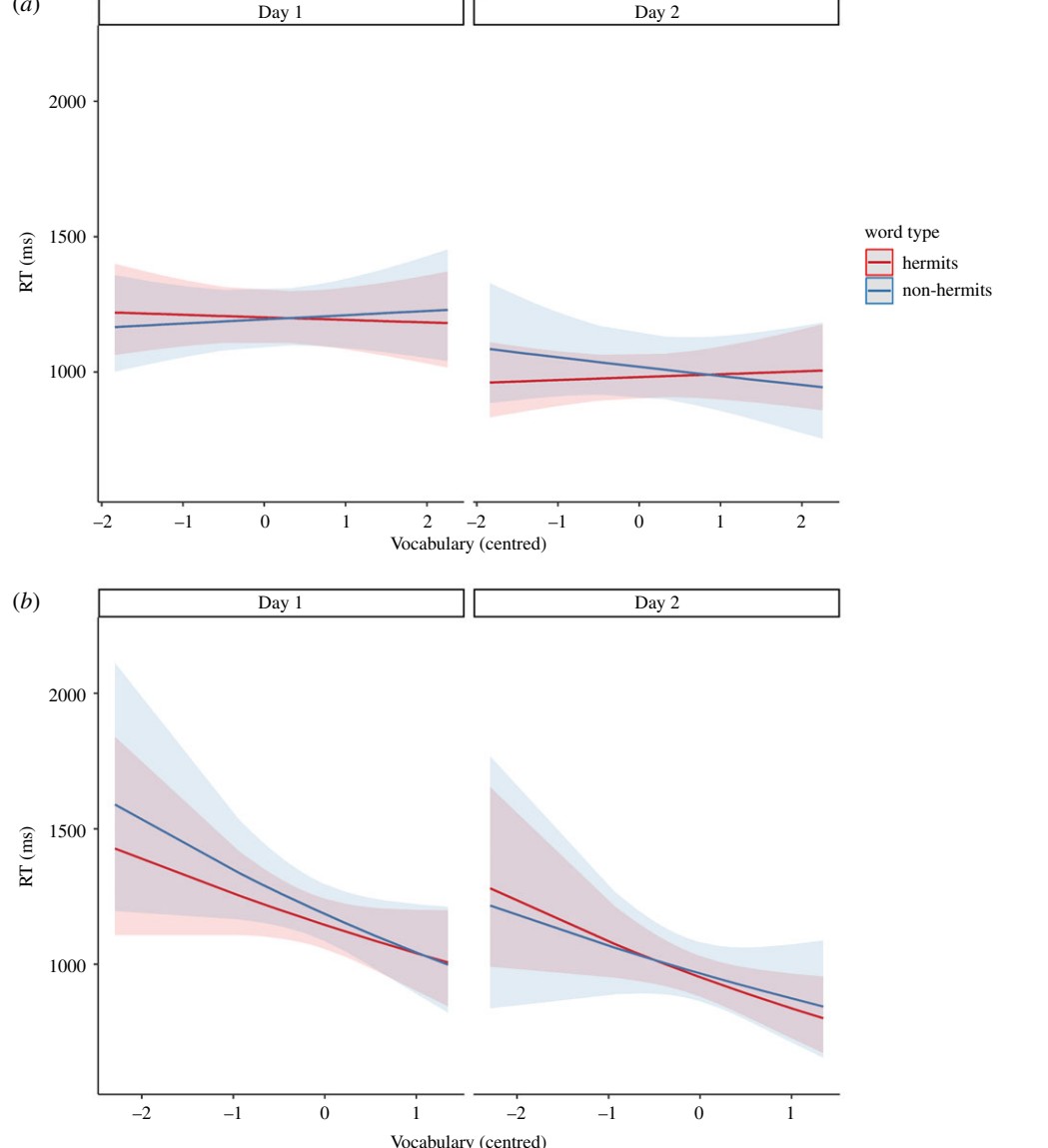

**Figure 2.** Semantic categorization back-transformed RT (ms) correlated with Vocabulary (scaled and centred) for both Day 1 and Day 2 for (*a*) the short delay group and (*b*) the long delay group.

**Table 2.** Mean (and s.d.) performance for the speeded recognition and cued recall tasks.

| delay group | day | speeded recognition (ms) | speeded recognition (% correct) | cued recall (% correct) |
|---|---|---|---|---|
| short | 1 | 1013 (242) | 96.1 (5.3) | 47.8 (27.0) |
|  | 2 | 911 (158) | 95.5 (6.1) | 52.7 (27.5) |
| long | 1 | 916 (219) | 98.1 (3.9) | 45.0 (31.1) |
|  | 2 | 910 (210) | 94.4 (6.4) | 45.2 (31.4) |

### 3.2.1. Speeded recognition

Inspection of table 2 suggests that participants in the short delay condition recognized the novel words more quickly after a consolidation opportunity (−102 ms), whereas participants in the long delay group did not (−6 ms). However, there was no main effect of Day ($b = -0.02$, s.e. = 0.03, $p = 0.49$), nor Vocabulary × Day ($b = -0.05$, s.e. = 0.04, $p = 0.26$) or Delay × Day interactions ($b = -0.07$, s.e. = 0.05,

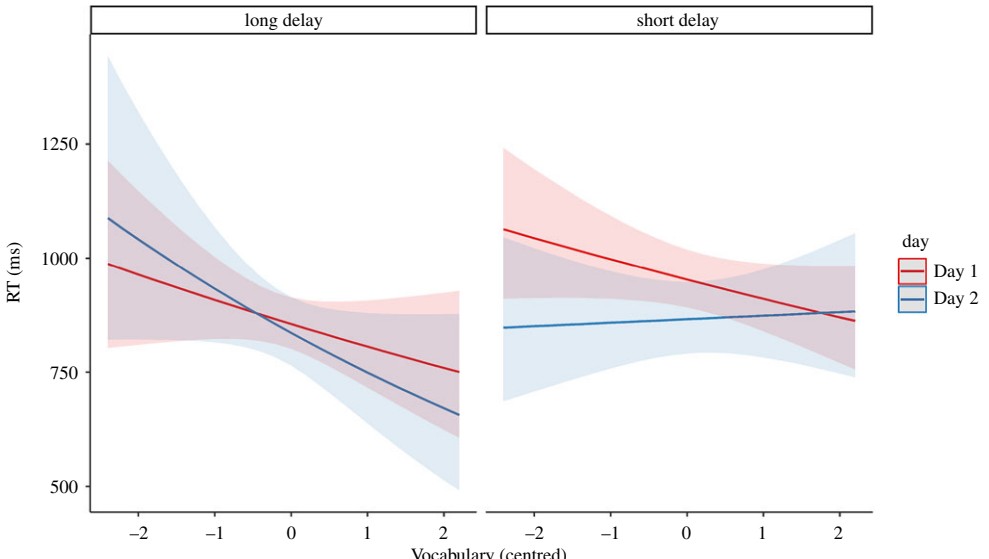

**Figure 3.** Speeded recognition backwards transformed RTs as a function of vocabulary (scaled and centred) and day for the long and short delay groups.

$p = 0.13$). Instead, there was a Vocabulary × Day × Delay interaction ($b = 0.10$, s.e. $= 0.05$, $p = 0.0495$). This effect appeared to be driven by two opposing trends depending on delay (figure 3). For both groups, performance on Day 1 provided a baseline, with participants scoring more highly on vocabulary showing faster response times. For the long delay group, this trend remained on Day 2, but for the short delay group, this trend was eliminated. Comparisons using emmeans found that neither of these individual trends was significant on its own (long delay group: $b = 0.05$, s.e. $= 0.05$, $p = 0.28$); short delay group: $b = -0.05$, s.e. $= 0.03$, $p = 0.0659$), implying that the significant three-way interaction reflected the combination of patterns.

### 3.2.2. Cued recall

Contrary to our predictions, there was no main effect of Day ($b = 0.16$, s.e. $= 0.13$, $p = 0.20$), indicating no overnight change in accuracy scores. Removal of the Delay × Day, Vocabulary × Day and Delay × Vocabulary × Day fixed effects did not significantly affect model fit ($p > 0.2$). Thus, recall accuracy was not influenced by either Vocabulary or Delay.

## 4. Discussion

It is well established that new spoken vocabulary undergoes a period of strengthening and integration offline; however, few studies have examined these offline consolidation processes in the written modality and relatively little is known about the optimal conditions that elicit consolidation. This study set out to investigate two key variables previously claimed to moderate consolidation effects in word learning—the time between learning and sleep and vocabulary knowledge—on children's ability to learn and consolidate new written words.

### 4.1. Do children show consolidation benefits in orthographic word learning?

Following explicit training of new orthographic forms, children achieved near-ceiling levels of performance on a speeded recognition task which maintained the following day. Children also showed levels of cued recall (approx. 46%) immediately after training that were somewhat higher than seen in similar previous studies (e.g. the authors in [18,23] report initial spoken cued recall rates of approx. 30%), and which were again maintained the following day. Unlike previous studies of orthographic word learning in adults that have used similar training regimes [35] and previous studies of spoken word learning with children [18,23,42], we did not see evidence of overnight *improvements* for the sample as a whole. This is somewhat surprising, given previous reports of enhanced overnight improvements in explicit memory for new words for children relative to adults

[22,41]. However, while it is possible that the relatively strong explicit memory traces in this study relied less on sleep-associated consolidation [28], it is also possible that the lack of decline observed in the explicit memory tests here could still reflect that consolidation processes guarded against forgetting.

There was also no evidence of lexical integration (i.e. indexed by lexical competition) on the day after learning that held for the participant group as a whole, nor was there evidence for immediate lexical integration that occurred on the day of learning [57]. The null results observed here conflict with previous studies of spoken word learning in children, which have consistently reported significant lexical competition effects on the day after training [18,23,42]. Tamura et al. [24] also found evidence of lexical engagement (measured by the prime-lexicality effect) following orthographic word learning from story contexts in children. As discussed in the Introduction, however, the semantic categorization task used here arguably captures a more automatic instantiation of lexical competition than the task used by Tamura et al. Orthographic lexical competition, as measured by the semantic categorization task, requires swift and automatic access of the phonological code upon presentation of the visual form. Given evidence that automatic orthographic recognition processes are not adult-like until at least age 11 (e.g. [58]), it is conceivable that there is significant variability in the latency and strength of orthographic lexical competition effects even in response to highly familiar stimuli in school-aged children. Another reason why we did not see basic consolidation effects here could be because, numerically at least, there were differences in consolidation effects for the short and long delay conditions, which may have diluted our chances of seeing these overall effects.

## 4.2. Does learning closer to bedtime give rise to superior consolidation?

Numerically, data trends from all three tasks were in the direction of stronger consolidation effects for the short delay group than the long delay group (i.e. greater overnight increases in lexical competition, greater decrease in recognition speed and bigger improvement in cued recall accuracy). However, only when taking vocabulary into account, did we find preliminary support for a difference between the short and long delay groups, in keeping with our fourth, more exploratory, hypothesis. For the short delay group only, weaker vocabulary was associated with larger lexical competition effects on the day after training. Thus, for children with poorer vocabulary, there was greater evidence of lexical integration if training was completed closer to bedtime (when there was less time that elapsed between tests, and thus less opportunity for interference/forgetting). Moreover, while vocabulary exerted similar effects on recognition performance on Day 1 for both delay conditions (i.e. lower vocabulary associated with slower recognition), vocabulary exerted opposing effects on Day 2 (i.e. lower vocabulary associated with slower recognition for the long delay group, but no influence of vocabulary for the short delay group). Conceivably, poor vocabulary compromised novel word acquisition, and when there was a longer period of post-learning wake between tests (i.e. more opportunity for interference/forgetting), this disadvantage remained after a period of consolidation. However, when sleep occurred soon after learning (i.e. less opportunity for interference/forgetting), consolidation compensated for weaker encoding and permitted lexical integration. Therefore, these data are partially consistent with previous claims of improved retrieval after sleep when encoding occurs close to sleep onset [24,29,31,32,36], but suggest a more nuanced stance—that sleeping close to bedtime is particularly beneficial when encoding is compromised by impoverished vocabulary knowledge. Notably, this pattern diverges from a comparable adult study [35], which used very similar materials and procedures; here, there was tentative evidence of a benefit for a *longer* time between learning and sleep which emerged a week after training. Although requiring replication, given the preliminary nature of the present four-way interaction, this points to a potential developmental difference (i.e. with some children, particularly those with lower vocabulary, being more prone to effects of wake-based interference), which should be explored directly in future studies.

It is important to point out limitations regarding the delay condition in the present study. Notably, the difference between the two delay groups was not only between learning and sleep, but also in the time between the two tests. That is, the long delay group had more time for interference or forgetting between tests (i.e. approx. 24 h) in contrast with the short delay group who had less opportunity for interference or forgetting (i.e. approx. 18 h). Thus, to determine whether it is the time between learning and sleep that is the important factor for children with lower vocabulary, or the time between tests, a further condition would be needed in which the short delay group are tested after 24 h. Encouragingly, however, a spoken word learning study by James et al. [59] has also demonstrated that children with lower vocabulary show long-term benefits of learning in the pm (as opposed to in the am) on a cued recall task, when controlling for the time between tests in an am–pm design. A second

limitation is that while all children were tested at school on the day after training, the long delay group were trained and initially tested at school, whereas the short delay group were trained and initially tested at home. Indeed, a previous study observed larger effects of sleep on memory for familiar lists of words when the context was different at encoding and retrieval than compared to when the context remained the same [60]. Since we cannot rule out that this confound led to differential effects for high versus low vocabulary children, this clearly needs to be addressed in future research.

## 4.3. Does existing knowledge facilitate the consolidation of new vocabulary?

Contrary to predictions, there was no evidence that better vocabulary was associated with larger consolidation effects (counter to [3], but consistent with [41]). Indeed, children at the top end of the vocabulary spectrum showed the *least* evidence of overnight consolidation in most cases. As mentioned above, it has been argued that there is a 'sweet spot' for consolidation, with weaker memory traces sometimes preferred for consolidation leading to stronger sleep-related consolidation effects [28]. Consistent with this, James *et al.* [41] found that children show better memory for novel words (e.g. regby) that have at least one word-form neighbour (e.g. rugby) than words with no neighbours immediately after training. However, by one week later, the no-neighbour items were consolidated to a greater extent, thus closing the gap between words with and without neighbours. It was argued that consolidation processes prioritized items that were initially less well encoded and/or less well connected to existing lexical knowledge. Such an explanation could therefore explain the lack of overnight sleep-related effects for those with higher levels of prior vocabulary knowledge here. It is worth highlighting that although children with better vocabulary scores did not show overnight improvements in explicit memory when encoding in the evening (in contrast with children with weaker vocabulary), neither did they show forgetting. It is possible therefore that sleep provides a more stabilizing function for children who are more likely to encode well. However, for children who are more likely to struggle at encoding (and have weaker memory traces), sleep may function to enhance their learning and/or memory (but only *if* learning occurs close enough to sleep).

## 5. Conclusion

This study examined whether it is optimal to maximize time awake following children's learning of new written words, or whether minimal time between learning and bedtime is beneficial to longer-term consolidation. The data did not provide strong evidence of overall sleep-dependent benefits on an orthographic word learning task, regardless of when in the day encoding took place, at least when evaluating the participant group as a whole. However, when taking into account individual differences, for children with poorer levels of receptive vocabulary, learning closer to sleep appeared most beneficial on tasks of both lexical integration and explicit memory. From a theoretical perspective, these data highlight the importance of understanding the parameters that influence the long-term consolidation of new vocabulary, earmarking sleep-based processes and vocabulary knowledge as two key variables. From a practical perspective, the findings are worthy of further investigation since they may have implications for the timing of learning prior to sleep, suggesting that children with poorer vocabulary may benefit from learning (or at least revising) new material closer to bedtime.

Ethics. All experiments were approved by the University of York Psychology Ethics Committee.

Data accessibility. All data are available at: https://osf.io/ptu25/. The experimental stimuli and mixed-effects analyses have been uploaded as part of the electronic supplementary material.

Authors' contributions. S.W. contributed to the design of the study, carried out the data acquisition, analysis and interpretation of the data, and drafted the manuscript; L.M.H. contributed to the conception, design, analysis and interpretation of the data, and helped draft the manuscript; F.E.F. contributed to the design, analysis and interpretation of the data, and helped draft the manuscript; V.C.P.K. contributed to the design of the study, the interpretation of the data and helped draft the manuscript; S.A.C. contributed to the conception and design of the study, and helped draft the manuscript; M.G.G. contributed to the conception and design of the study, the analysis and interpretation of the data, and helped draft the manuscript. All authors gave final approval for publication.

Competing interests. We declare we have no competing interests.

Funding. The research was supported by UK Economic and Social Research Council grant no. ES/N009924/1 (awarded to L.M.H., M.G.G. and Courtenay Norbury).

Acknowledgements. First, we thank the participants for taking part in this research. We are very grateful to members of the Sleep, Language and Memory Lab for valuable discussion of this work.

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
