## [Reviewer comments · Royal Society Open Science]

Review History

RSOS-191597.R0 (Original submission)

Review form: Reviewer 1

Is the manuscript scientifically sound in its present form?

Yes

Are the interpretations and conclusions justified by the results?

No

Is the language acceptable?

Yes

Do you have any ethical concerns with this paper?

No

Have you any concerns about statistical analyses in this paper?

No

Recommendation?

Major revision is needed (please make suggestions in comments)

Comments to the Author(s)

Summary:

The article presents one experiment investigating novel word learning by children. The authors aimed at examining the effects of 1) sleep, 2) delay between learning and bedtime, and 3) vocabulary size, as well as the interactions between these variables, on 1) lexical integration and 2) explicit memory of the novel items. Lexical integration was measured as the difference between hermit and non-hermit words, under the rationale that when a hermit word (e.g., BANANA) gains a neighbor, its recognition should be slowed down due to competition from its new neighbor.

In regard to lexical integration, a significant 4-way Vocabulary x Delay x Day x Word type interaction was found. As for explicit memory, a significant 3-way Vocabulary x Day x Delay interaction is reported.

Disclaimer:

I reviewed another version of this paper submitted to a different journal (May 2019) . This is a modified and improved version of the earlier version I reviewed and I was happy to see that some of my comments/suggestions are now addressed. That said, I am afraid that the core issues -regarding the analytical approach and interpretation of the results- remain. This is why, as the authors will see, many of my comments remain the same from last time.

Review:

This is a very nice study that addresses an interesting question. The literature review is quite thorough, the hypothesis and predictions are clearly spelled-out, and the experimental methods are laid down in a clear and well-organized manner. Overall, the writing is very clear and the narrative is easy to follow.

Introduction/Literature review:

The literature review is quite thorough and I was happy to see that the authors now acknowledge the accumulating evidence for sleep-less lexical integration, as well as immediate lexical integration. Including reference to this work helps provide the readers with a more balanced view of the related research.

Results and Major concern:

In regard to lexical integration, the authors report that neither the Word type x Day interaction, nor either one of the two three-way interactions (between Word type x Day x Vocabulary and Word type x Day x Delay) were significant. The only significant result in this first analysis was the 4-way Vocabulary x Delay x Day x Word type interaction. The emmeans R package was used to further examine this 4-way interaction. This second analysis showed that "For the Short Delay group, the slopes for the different word type were more variable, with a significant difference emerging on Day 2 but not Day 1 [...]". The authors argue that this result can be taken as "tentative evidence that lexical competition effects were emerging on Day 2 for the Short Delay group, particularly for children with lower vocabulary". One issue here is that, according to information provided in a previous version of the manuscript (see Disclaimer above), this Day 2 effect does not in fact survive correction for multiple comparisons. If this is still the case, the authors may want to add this information back into the manuscript and adjust their conclusions accordingly.

Putting aside the issue of multiple comparisons, what to me is the most problematic aspect of this (in many other respects lovely) work is that the analytical approach taken seems to be quite counterintuitive when it comes to evaluating the authors' hypotheses. That is, given the theoretical framing of the study, the critical effect of interest is that of Word type (i.e., the difference between hermits and non-hermits), which reflects lexical integration. Furthermore, I

understand that the main aims of the study were to examine how lexical integration is affected by sleep (Day \times Word type interaction; see aim #1: “We hypothesized that children would show overnight increases in lexical competition between novel and existing competitors, and overnight improvements in their ability to recognise and recall the novel words”) and how this effect of sleep (i.e., consolidation) is modulated by 1) the delay between learning and sleep (Word type \times Day \times Delay; see aim #2: “Second, we examined whether the time between learning and sleep influences consolidation, with the prediction that learning closer to sleep would give rise to greater improvements in overnight consolidation in childhood [...]”) and 2) vocabulary knowledge (Word type \times Day \times Vocabulary interaction; see aim #3: “Third, we predicted that overnight increases in lexical competition would be associated with better vocabulary knowledge based on claims that existing knowledge facilitates lexical integration [...]”). Lastly, the authors were also interested in the interplay between these two factors (delay and vocabulary) in affecting consolidation (Word type \times Day \times Delay \times Vocabulary; see: “A further exploratory hypothesis [...]”). In other words, the other effects (sleep, delay, and vocabulary) are examined in respect to how they modulate the former main effect of interest (word type). In contrast, the current analytical approach seems to be focusing on the Vocabulary effect, rather than the Word type effect (i.e., see p.15, lines 35-40: “This interaction [...] was explored further using the R package emmeans [56] to examine the effect of vocabulary in each cell of the design [...]”). As a result, it is difficult to interpret the results in relation to the hypotheses presented in the “Present study” section; the fact that vocabulary may have a significantly different simple effect for hermit versus non-hermit items on Day 2 for the Short delay group doesn’t really directly address any of the hypotheses.

Given the framing of the predictions, it seems that a more appropriate analytical approach would focus on lexical integration (Word Type effect) and consolidation (Day \times Word type interaction). For example, one could examine the 4-way interaction as following: first, split the data by Delay Group to check which 3-way (Word type \times Day \times Vocabulary) interaction is significant (i.e., for each of the two delay groups separately). Then, follow up on the significant 3-way interaction(s) by splitting again by Vocabulary score (e.g., via a median split). Then, at the end, looking at which 2-way interaction(s) is/are significant would be very informative; if, for example, for a given group there is a significant Word type \times Day interaction but only for children with low vocabulary score, this would be quite meaningful. For example, given the pattern shown in Figure 2, this could mean that children with weaker vocabulary in the lower panel group show evidence for lexical integration immediately after training, on Day 1, but not on Day 2. In contrast, for the children with weaker vocabulary in the upper panel group, lexical integration only appears on Day 2, which may indicate a delay in regard to when lexical integration effects first appear.

Additional suggestions:

- 1) Performance on Day 1 is consistently used as a baseline. I understand why this is the case, given that the authors are interested in sleep-driven consolidation. However, given a) previous literature showing evidence for lexical integration immediately after training, and b) the absence of evidence for consolidation in this study, the authors may want to consider acknowledging that Day 1 performance may in fact reflect early stages of lexical integration.
- 2) A potentially interesting analysis would be to extract a measure of lexical integration (i.e., Non-hermit RT - Hermit RT) for each child on Day 1 and Day 2 and test whether Day 1 score predicts Day 2 score. This type of analysis would allow one to see whether immediate/early integration is a predictor of post-sleep integration and whether this relationship is modulated by Delay and/or Vocabulary.

Minor suggestions:

1. On p. 15 (lines 14-15), it is reported that “RTs were faster overall on Day 2 (M = 1055 ms, SD = 179) than Day 1 (M = 1280 ms, SD = 192)”. It would be helpful to include the stats here, given that the difference was significant.
2. On p. 15, the authors mention that “For the Long Delay group, there was a general trend of faster responses for participants with higher vocabulary scores [...]”. Again, this might be a good point to include the statistical output for the 2-way Delay \times Vocabulary interaction.

Recommendation:

Despite this being a very interesting and well-conducted study, I am afraid I cannot recommend it for publication in its current form. A number of concerns and suggestions are mentioned above, among which I can point to two major issues: 1) Given that one of the effects does not survive correction for multiple comparisons, it would be appropriate to tone down the conclusions based on this result and instead focus on the significant results. 2) I strongly suggest following-up on the 4-way (Vocabulary × Delay × Day × Word type) interaction in a different way, so that the results can be better linked to the hypotheses.

Review form: Reviewer 2 (Padraic Monaghan)**Is the manuscript scientifically sound in its present form?**

Yes

Are the interpretations and conclusions justified by the results?

Yes

Is the language acceptable?

Yes

Do you have any ethical concerns with this paper?

No

Have you any concerns about statistical analyses in this paper?

Yes

Recommendation?

Accept with minor revision (please list in comments)

Comments to the Author(s)

This is a really interesting study, investigating the relation between learning new words and sleep in 8 to 12 year old children, when the time interval between a study phase and sleep (during which consolidation of learning can take place) varies. The study tests an under-studied group investigating effects of learning and memory that have potentially important practical implications for pedagogy, as well as theoretical models of sleep and memory. The results failed to replicate previous studies on the consolidation effects of sleep, indicating no overall difference on lexical integration between immediate versus delayed sleep. However, explorative analyses demonstrated a significant interaction between immediate versus delayed sleep and vocabulary size. Children with smaller vocabularies benefited more from immediate sleep than delayed sleep in lexical integration. For children with larger vocabularies, there was no evidence difference between immediate and delayed sleep.

The lack of replication of previous studies on consolidation effects associated with sleep are effectively discussed in the paper, and the writing and presentation of the work is exemplary in its detail, reflective quality, and integration with the literature. The work is a useful addition to the literature, though the significant results in the paper represent small effects in complex interactions, and may be difficult to replicate.

The authors mention on p.9 that the four-way interactions between vocabulary size, delay, day, and item type is an explorative hypothesis, and the interaction just reaches a conventional level of

significance. This is ok – that's what conventions are for – but there are also a lot of filtering of outliers performed before the analyses are performed (pp.13-14). I would like the authors to also mention whether the results are robust to inclusion of outliers, so that the robustness of the four-way interaction can be ascertained. It is of course justified to remove outliers, but if the results change when some of these outliers are included, then that is useful additional knowledge, and does not affect the main message of the paper.

Relatedly, can the thresholds for identifying outliers be justified? On p.13 participants with z-scores larger in magnitude than $-/+3.29$ are removed. Why this value? On p.14 participants with z-scores less than -2.5 were removed. Why this value? On p.14 participants with BPVS z-scores greater than 2.5 were removed. Also it looks like scores < -2.5 were removed. Why this value?

Throughout, the authors should refer to explorative analyses for additional analyses that were not initially planned during the execution of the paper. For instance, hypothesis 3 is explorative. Is hypothesis 4 also explorative? It seems equally complex. When hypotheses are being discussed with respect to the results, if the explorative nature of the analysis can also be mentioned that again helps the reader to navigate the research.

Decision letter (RSOS-191597.R0)

22-Nov-2019

Dear Dr Walker,

The editors assigned to your paper ("Growing up with interfering neighbours: The influence of time of learning and vocabulary knowledge on written word learning in children") have now received comments from reviewers. We would like you to revise your paper in accordance with the referee and Associate Editor suggestions which can be found below (not including confidential reports to the Editor). Please note this decision does not guarantee eventual acceptance.

Please submit a copy of your revised paper before 15-Dec-2019. Please note that the revision deadline will expire at 00.00am on this date. If we do not hear from you within this time then it will be assumed that the paper has been withdrawn. In exceptional circumstances, extensions may be possible if agreed with the Editorial Office in advance. We do not allow multiple rounds of revision so we urge you to make every effort to fully address all of the comments at this stage. If deemed necessary by the Editors, your manuscript will be sent back to one or more of the original reviewers for assessment. If the original reviewers are not available, we may invite new reviewers.

- Data accessibility

If you wish to submit your supporting data or code to Dryad (<http://datadryad.org/>), or modify your current submission to dryad, please use the following link:
<http://datadryad.org/submit?journalID=RSOS&manu=RSOS-191597>

- Competing interests

- Authors' contributions

- Acknowledgements

- Funding statement

Kind regards,

on behalf of Dr Emma Hayiou-Thomas (Associate Editor) and Essi Viding (Subject Editor)
openscience@royalsociety.org

Associate Editor's comments (Dr Emma Hayiou-Thomas):

I enjoyed reading this paper, and agree with both reviewers that this is a well-designed study addressing an interesting topic. The lack of the predicted main effects of overnight consolidation is surprising, but the interaction you report between time of learning and existing vocabulary is interesting, and one that I hope will be followed up in future studies. Both reviewers make constructive suggestions which will need to be addressed, several of which involve further analyses. For example, Reviewer 2 recommends that you report whether the results are materially changed by the inclusion of outliers. However, the most substantial point that a revision will need to address is the exploration of the 4-way interaction. Reviewer 2 strongly recommends that in order to directly test your hypotheses, you should carry out a series of follow-up analyses, along the lines that she has very clearly laid out. I agree that this would aid in the interpretation of the 4-way interaction, and further strengthen the paper.

Reviewers' Comments to Author:

Reviewer: 1
Comments to the Author(s)

Summary:

The article presents one experiment investigating novel word learning by children. The authors aimed at examining the effects of 1) sleep, 2) delay between learning and bedtime, and 3) vocabulary size, as well as the interactions between these variables, on 1) lexical integration and 2) explicit memory of the novel items. Lexical integration was measured as the difference between hermit and non-hermit words, under the rationale that when a hermit word (e.g., BANANA) gains a neighbor, its recognition should be slowed down due to competition from its new neighbor.

In regard to lexical integration, a significant 4-way Vocabulary x Delay x Day x Word type interaction was found. As for explicit memory, a significant 3-way Vocabulary x Day x Delay interaction is reported.

Disclaimer:

I reviewed another version of this paper submitted to a different journal (May 2019). This is a modified and improved version of the earlier version I reviewed and I was happy to see that some of my comments/suggestions are now addressed. That said, I am afraid that the core issues -regarding the analytical approach and interpretation of the results- remain. This is why, as the authors will see, many of my comments remain the same from last time.

Review:

This is a very nice study that addresses an interesting question. The literature review is quite thorough, the hypothesis and predictions are clearly spelled-out, and the experimental methods

are laid down in a clear and well-organized manner. Overall, the writing is very clear and the narrative is easy to follow.

Introduction/Literature review:

The literature review is quite thorough and I was happy to see that the authors now acknowledge the accumulating evidence for sleep-less lexical integration, as well as immediate lexical integration. Including reference to this work helps provide the readers with a more balanced view of the related research.

Results and Major concern:

In regard to lexical integration, the authors report that neither the Word type \times Day interaction, nor either one of the two three-way interactions (between Word type \times Day \times Vocabulary and Word type \times Day \times Delay) were significant. The only significant result in this first analysis was the 4-way Vocabulary \times Delay \times Day \times Word type interaction. The emmeans R package was used to further examine this 4-way interaction. This second analysis showed that “For the Short Delay group, the slopes for the different word type were more variable, with a significant difference emerging on Day 2 but not Day 1 [...]”. The authors argue that this result can be taken as “tentative evidence that lexical competition effects were emerging on Day 2 for the Short Delay group, particularly for children with lower vocabulary”. One issue here is that, according to information provided in a previous version of the manuscript (see Disclaimer above), this Day 2 effect does not in fact survive correction for multiple comparisons. If this is still the case, the authors may want to add this information back into the manuscript and adjust their conclusions accordingly.

Putting aside the issue of multiple comparisons, what to me is the most problematic aspect of this (in many other respects lovely) work is that the analytical approach taken seems to be quite counterintuitive when it comes to evaluating the authors’ hypotheses. That is, given the theoretical framing of the study, the critical effect of interest is that of Word type (i.e., the difference between hermits and non-hermits), which reflects lexical integration. Furthermore, I understand that the main aims of the study were to examine how lexical integration is affected by sleep (Day \times Word type interaction; see aim #1: “We hypothesized that children would show overnight increases in lexical competition between novel and existing competitors, and overnight improvements in their ability to recognise and recall the novel words”) and how this effect of sleep (i.e., consolidation) is modulated by 1) the delay between learning and sleep (Word type \times Day \times Delay; see aim #2: “Second, we examined whether the time between learning and sleep influences consolidation, with the prediction that learning closer to sleep would give rise to greater improvements in overnight consolidation in childhood [...]”) and 2) vocabulary knowledge (Word type \times Day \times Vocabulary interaction; see aim #3: “Third, we predicted that overnight increases in lexical competition would be associated with better vocabulary knowledge based on claims that existing knowledge facilitates lexical integration [...]”). Lastly, the authors were also interested in the interplay between these two factors (delay and vocabulary) in affecting consolidation (Word type \times Day \times Delay \times Vocabulary; see: “A further exploratory hypothesis [...]”). In other words, the other effects (sleep, delay, and vocabulary) are examined in respect to how they modulate the former main effect of interest (word type). In contrast, the current analytical approach seems to be focusing on the Vocabulary effect, rather than the Word type effect (i.e., see p.15, lines 35-40: “This interaction [...] was explored further using the R package emmeans [56] to examine the effect of vocabulary in each cell of the design [...]”). As a result, it is difficult to interpret the results in relation to the hypotheses presented in the “Present study” section; the fact that vocabulary may have a significantly different simple effect for hermit versus non-hermit items on Day 2 for the Short delay group doesn’t really directly address any of the hypotheses.

Given the framing of the predictions, it seems that a more appropriate analytical approach would focus on lexical integration (Word Type effect) and consolidation (Day \times Word type interaction). For example, one could examine the 4-way interaction as following: first, split the data by Delay Group to check which 3-way (Word type \times Day \times Vocabulary) interaction is significant (i.e., for each of the two delay groups separately). Then, follow up on the significant 3-way interaction(s)

by splitting again by Vocabulary score (e.g., via a median split). Then, at the end, looking at which 2-way interaction(s) is/are significant would be very informative; if, for example, for a given group there is a significant Word type \times Day interaction but only for children with low vocabulary score, this would be quite meaningful. For example, given the pattern shown in Figure 2, this could mean that children with weaker vocabulary in the lower panel group show evidence for lexical integration immediately after training, on Day 1, but not on Day 2. In contrast, for the children with weaker vocabulary in the upper panel group, lexical integration only appears on Day 2, which may indicate a delay in regard to when lexical integration effects first appear.

Additional suggestions:

- 1) Performance on Day 1 is consistently used as a baseline. I understand why this is the case, given that the authors are interested in sleep-driven consolidation. However, given a) previous literature showing evidence for lexical integration immediately after training, and b) the absence of evidence for consolidation in this study, the authors may want to consider acknowledging that Day 1 performance may in fact reflect early stages of lexical integration.
- 2) A potentially interesting analysis would be to extract a measure of lexical integration (i.e., Non-hermit RT - Hermit RT) for each child on Day 1 and Day 2 and test whether Day 1 score predicts Day 2 score. This type of analysis would allow one to see whether immediate/early integration is a predictor of post-sleep integration and whether this relationship is modulated by Delay and/or Vocabulary.

Minor suggestions:

1. On p. 15 (lines 14-15), it is reported that "RTs were faster overall on Day 2 ($M = 1055$ ms, $SD = 179$) than Day 1 ($M = 1280$ ms, $SD = 192$)". It would be helpful to include the stats here, given that the difference was significant.
2. On p. 15, the authors mention that "For the Long Delay group, there was a general trend of faster responses for participants with higher vocabulary scores [...]". Again, this might be a good point to include the statistical output for the 2-way Delay \times Vocabulary interaction.

Recommendation:

Despite this being a very interesting and well-conducted study, I am afraid I cannot recommend it for publication in its current form. A number of concerns and suggestions are mentioned above, among which I can point to two major issues: 1) Given that one of the effects does not survive correction for multiple comparisons, it would be appropriate to tone down the conclusions based on this result and instead focus on the significant results. 2) I strongly suggest following-up on the 4-way (Vocabulary \times Delay \times Day \times Word type) interaction in a different way, so that the results can be better linked to the hypotheses.

Reviewer: 2

Comments to the Author(s)

This is a really interesting study, investigating the relation between learning new words and sleep in 8 to 12 year old children, when the time interval between a study phase and sleep (during which consolidation of learning can take place) varies. The study tests an under-studied group investigating effects of learning and memory that have potentially important practical implications for pedagogy, as well as theoretical models of sleep and memory. The results failed to replicate previous studies on the consolidation effects of sleep, indicating no overall difference on lexical integration between immediate versus delayed sleep. However, explorative analyses demonstrated a significant interaction between immediate versus delayed sleep and vocabulary size. Children with smaller vocabularies benefited more from immediate sleep than delayed sleep in lexical integration. For children with larger vocabularies, there was no evidence difference between immediate and delayed sleep.

The lack of replication of previous studies on consolidation effects associated with sleep are effectively discussed in the paper, and the writing and presentation of the work is exemplary in its detail, reflective quality, and integration with the literature. The work is a useful addition to the literature, though the significant results in the paper represent small effects in complex interactions, and may be difficult to replicate.

The authors mention on p.9 that the four-way interactions between vocabulary size, delay, day, and item type is an explorative hypothesis, and the interaction just reaches a conventional level of significance. This is ok – that’s what conventions are for – but there are also a lot of filtering of outliers performed before the analyses are performed (pp.13-14). I would like the authors to also mention whether the results are robust to inclusion of outliers, so that the robustness of the four-way interaction can be ascertained. It is of course justified to remove outliers, but if the results change when some of these outliers are included, then that is useful additional knowledge, and does not affect the main message of the paper.

Relatedly, can the thresholds for identifying outliers be justified? On p.13 participants with z-scores larger in magnitude than $-/+3.29$ are removed. Why this value? On p.14 participants with z-scores less than -2.5 were removed. Why this value? On p.14 participants with BPVS z-scores greater than 2.5 were removed. Also it looks like scores < -2.5 were removed. Why this value?

Throughout, the authors should refer to explorative analyses for additional analyses that were not initially planned during the execution of the paper. For instance, hypothesis 3 is explorative. Is hypothesis 4 also explorative? It seems equally complex. When hypotheses are being discussed with respect to the results, if the explorative nature of the analysis can also be mentioned that again helps the reader to navigate the research.

Author's Response to Decision Letter for (RSOS-191597.R0)

See Appendix A.

RSOS-191597.R1 (Revision)

Review form: Reviewer 2 (Padraic Monaghan)

Is the manuscript scientifically sound in its present form?

Yes

Are the interpretations and conclusions justified by the results?

Yes

Is the language acceptable?

Yes

Do you have any ethical concerns with this paper?

No

Have you any concerns about statistical analyses in this paper?

No

Recommendation?

Accept with minor revision (please list in comments)

Comments to the Author(s)

This is an excellent revision of the first version of the paper.

The authors have addressed my primary concern about the explorative analyses being clearly highlighted.

My other concern was about the identification of outliers. The authors have now taken two approaches - one based on *dfbetas*, and the other on identifying *z*scores for participants. These are both now mentioned on pp. of the paper, but it is not clear to me the order in which these were adopted (or even whether the omission of outliers using the *z*scores is superfluous given the use of *dfbetas*). In a final adjustment, please can the authors clarify the relation between these outlier analyses (the order in which they are applied, and why they are both still necessary)? It seems that the *dfbetas* analysis removes participants, whereas the *z*scores approach removes individual observations from participants (that at least is what the description on p.13 seems to suggest). The authors suggest this is more conservative, so is the removal via *z*scores of individual participant's responses still necessary? Stating this practice explicitly will be helpful in future studies that aim to use similar models.

Decision letter (RSOS-191597.R1)

21-Feb-2020

Dear Dr Walker:

On behalf of the Editors, I am pleased to inform you that your Manuscript RSOS-191597.R1 entitled "Growing up with interfering neighbours: The influence of time of learning and vocabulary knowledge on written word learning in children" has been accepted for publication in Royal Society Open Science subject to minor revision in accordance with the referee suggestions. Please find the referees' comments at the end of this email.

The reviewers and Subject Editor have recommended publication, but also suggest some minor revisions to your manuscript. Therefore, I invite you to respond to the comments and revise your manuscript.

- Ethics statement

- Data accessibility

If you wish to submit your supporting data or code to Dryad (<http://datadryad.org/>), or modify your current submission to dryad, please use the following link:
<http://datadryad.org/submit?journalID=RSOS&manu=RSOS-191597.R1>

- **Competing interests**

- **Authors' contributions**

- **Acknowledgements**

- **Funding statement**

Because the schedule for publication is very tight, it is a condition of publication that you submit the revised version of your manuscript before 01-Mar-2020. Please note that the revision deadline will expire at 00.00am on this date. If you do not think you will be able to meet this date please let me know immediately.

on behalf of Prof Essi Viding (Subject Editor)
openscience@royalsociety.org

Editor Comments to Author (Dr Essi Viding):

Please address the final minor point raised by reviewer re: outlier analysis.

Reviewer comments to Author:
Reviewer: 2

Comments to the Author(s)
This is an excellent revision of the first version of the paper.

The authors have addressed my primary concern about the explorative analyses being clearly highlighted.

My other concern was about the identification of outliers. The authors have now taken two approaches - one based on dfbetas, and the other on identifying zscores for participants. These are both now mentioned on pp. of the paper, but it is not clear to me the order in which these were adopted (or even whether the omission of outliers using the zscores is superfluous given the use of dfbetas). In a final adjustment, please can the authors clarify the relation between these

outlier analyses (the order in which they are applied, and why they are both still necessary)? It seems that the dfbetas analysis removes participants, whereas the zscores approach removes individual observations from participants (that at least is what the description on p.13 seems to suggest). The authors suggest this is more conservative, so is the removal via zscores of individual participant's responses still necessary? Stating this practice explicitly will be helpful in future studies that aim to use similar models.

Author's Response to Decision Letter for (RSOS-191597.R1)

See Appendix B.

Decision letter (RSOS-191597.R2)

27-Feb-2020

Dear Dr Walker,

It is a pleasure to accept your manuscript entitled "Growing up with interfering neighbours: The influence of time of learning and vocabulary knowledge on written word learning in children" in its current form for publication in Royal Society Open Science. The comments of the reviewer(s) who reviewed your manuscript are included at the foot of this letter.

on behalf of Prof Essi Viding (Subject Editor)
openscience@royalsociety.org

Appendix A

3rd December 2019

Dear Editors,

Re: Growing up with interfering neighbours: The influence of time of learning and vocabulary knowledge on written word learning in children

Thank you for the detailed feedback on the above paper. We have rewritten the paper in the light of these valuable comments and feel that the manuscript is now substantially improved. We hope that you agree that we have been able to satisfactorily address the comments or provide justification where an alternative approach is taken, and that you feel the manuscript is now suitable for publication in *Royal Society Open Science*. Of course, we are very happy to respond to any further comments. The details of how we have addressed the specific issues raised are below.

There is just one issue we wish to raise. As you are aware, there is a potential conflict of interest here, given the assigned Action Editor works in the same department as the authors of this paper. Whilst we are very confident that this would not influence the review process or the final decision on the manuscript in any way, we are slightly concerned that it could reflect badly on the outside. Thus, we would prefer it if the final decision making was at least given the stamp of approval by another Editor.

We look forward to hearing from you in due course.

Yours sincerely,

Dr Sarah Walker

Reviewer #1

I reviewed another version of this paper submitted to a different journal (May 2019) . This is a modified and improved version of the earlier version I reviewed and I was happy to see that some of my comments/suggestions are now addressed. That said, I am afraid that the core issues -regarding the analytical approach and interpretation of the results- remain. This is why, as the authors will see, many of my comments remain the same from last time.

This is a very nice study that addresses an interesting question. The literature review is quite thorough, the hypothesis and predictions are clearly spelled-out, and the experimental methods are laid down in a clear and well-organized manner. Overall, the writing is very clear and the narrative is easy to follow.

In regard to lexical integration, the authors report that neither the Word type \times Day interaction, nor either one of the two three-way interactions (between Word type \times Day \times Vocabulary and Word type \times Day \times Delay) were significant. The only significant result in this first analysis was the 4-way Vocabulary \times Delay \times Day \times Word type interaction. The emmeans R package was used to further examine this 4-way interaction. This second analysis showed that “For the Short Delay group, the slopes for the different word type were more variable, with a significant difference emerging on Day 2 but not Day 1 [...]”. The authors argue that this result can be taken as “tentative evidence that lexical competition effects were emerging on Day 2 for the Short Delay group, particularly for children with lower vocabulary”. One issue here is that, according to information provided in a previous version of the manuscript (see Disclaimer above), this Day 2 effect does not in fact survive correction for multiple comparisons. If this is still the case, the authors may want to add this information back into the manuscript and adjust their conclusions accordingly.

Upon re-reading and resubmitting our manuscript, we felt as though it was sufficient to be very clear about the fact that the four way interaction is not strong and is tentative evidence at best. It's clear that the p value is .04 and would not survive any form of correction of the number of main effects/interactions. That said, we have added in that the p value for this interaction would not withstand correction “Although the strength of the statistical evidence here is slight (and it should be noted that the four way interaction would not survive correction for multiple comparisons), the interaction provides tentative evidence that lexical competition effects were emerging on Day 2 for the Short Delay group, particularly for children with lower vocabulary.” (p. 16). Importantly, we have toned down our conclusions in the Discussion to emphasize the preliminary nature of the evidence and the exploratory nature of the hypothesis, for example, when we introduce our discussion of the 4 way interaction we state “However, only when taking vocabulary into account did we find preliminary support a difference between Short and Long Delay groups, in keeping with our fourth more exploratory hypothesis.” We have also reworded the final paragraph.

Putting aside the issue of multiple comparisons, what to me is the most problematic aspect of this (in many other respects lovely) work is that the analytical approach taken seems to be quite counterintuitive when it comes to evaluating the authors' hypotheses. That is, given the

theoretical framing of the study, the critical effect of interest is that of Word type (i.e., the difference between hermits and non-hermits), which reflects lexical integration. Furthermore, I understand that the main aims of the study were to examine how lexical integration is affected by sleep (Day \times Word type interaction; see aim #1: “We hypothesized that children would show overnight increases in lexical competition between novel and existing competitors, and overnight improvements in their ability to recognise and recall the novel words”) and how this effect of sleep (i.e., consolidation) is modulated by 1) the delay between learning and sleep (Word type \times Day \times Delay; see aim #2: “Second, we examined whether the time between learning and sleep influences consolidation, with the prediction that learning closer to sleep would give rise to greater improvements in overnight consolidation in childhood [...]”) and 2) vocabulary knowledge (Word type \times Day \times Vocabulary interaction; see aim #3: “Third, we predicted that overnight increases in lexical competition would be associated with better vocabulary knowledge based on claims that existing knowledge facilitates lexical integration [...]”). Lastly, the authors were also interested in the interplay between these two factors (delay and vocabulary) in affecting consolidation (Word type \times Day \times Delay \times Vocabulary; see: “A further exploratory hypothesis [...]”). In other words, the other effects (sleep, delay, and vocabulary) are examined in respect to how they modulate the former main effect of interest (word type). In contrast, the current analytical approach seems to be focusing on the Vocabulary effect, rather than the Word type effect (i.e., see p.15, lines 35-40: “This interaction [...] was explored further using the R package emmeans [56] to examine the effect of vocabulary in each cell of the design [...]”). As a result, it is difficult to interpret the results in relation to the hypotheses presented in the “Present study” section; the fact that vocabulary may have a significantly different simple effect for hermit versus non-hermit items on Day 2 for the Short delay group doesn’t really directly address any of the hypotheses.

Given the framing of the predictions, it seems that a more appropriate analytical approach would focus on lexical integration (Word Type effect) and consolidation (Day \times Word type interaction). For example, one could examine the 4-way interaction as following: first, split the data by Delay Group to check which 3-way (Word type \times Day \times Vocabulary) interaction is significant (i.e., for each of the two delay groups separately). Then, follow up on the significant 3-way interaction(s) by splitting again by Vocabulary score (e.g., via a median split). Then, at the end, looking at which 2-way interaction(s) is/are significant would be very informative; if, for example, for a given group there is a significant Word type \times Day interaction but only for children with low vocabulary score, this would be quite meaningful. For example, given the pattern shown in Figure 2, this could mean that children with weaker vocabulary in the lower panel group show evidence for lexical integration immediately after training, on Day 1, but not on Day 2. In contrast, for the children with weaker vocabulary in the upper panel group, lexical integration only appears on Day 2, which may indicate a delay in regard to when lexical integration effects first appear.

We thank the reviewer for these comments. We did consider including the suggested analyses in our initial submission to RSOS, but we had a number of concerns about them. One is that we did not wish to depart from the original model in running further analyses of subsets of

data. As the FAQ for emmeans states, “Doing separate analyses on subsets usually comprises departing from that overall model, so of course the results are different.” The sub-analyses we report are all based on this full model and so interrogate the 4-way interaction in its purest form. The second concern was that a median split has its own well-documented weaknesses. Furthermore, the 4-way interaction, as we stated, is not a strong statistical basis for substantial further analyses, and we wanted to take a lighter touch approach compared with the reviewer recommendations.

All that said, we agree that the current analyses and the graphs in Figure 2 are quite hard to interpret. Therefore we suggest that we include a simpler form of the suggested analyses as a footnote. The footnote reads:

Based on a reviewer’s suggestion, we also interrogated the 4-way interaction by running analyses on each of the delay groups separately. There was a significant Word type x Day x Vocabulary interaction in the Short delay group alone ($b = -.07$, $SE = .03$, $p = .025$). To follow this up we divided the Vocabulary variable using a median split. The Word type x Day interaction was approaching significance in the Low vocab ($b = -.29$, $SE = .15$, $p = .07$), but not the High vocab group ($b = .10$, $SE = .13$, $p = .44$).

Additional suggestions:

1) Performance on Day 1 is consistently used as a baseline. I understand why this is the case, given that the authors are interested in sleep-driven consolidation. However, given a) previous literature showing evidence for lexical integration immediately after training, and b) the absence of evidence for consolidation in this study, the authors may want to consider acknowledging that Day 1 performance may in fact reflect early stages of lexical integration.

Although this is a really interesting point, it is worth reminding that we did not see convincing evidence of lexical competition on Day 1, or on Day 1 and Day 2 combined. The only case in which we see statistical support for evidence of lexical competition is on Day 2 for children with weaker vocabulary, if they have learned in the evening. Nevertheless, we have added further acknowledgement in the Discussion that the results do not support the hypothesis that lexical competition emerges on day 2, but nor do they support the alternative hypothesis that lexical competition emerges immediately after learning: “There was also no evidence of lexical integration (i.e., indexed by lexical competition) on the day after learning that held for the participant group as a whole, nor was there evidence for immediate lexical integration that occurred on the day of learning (Kapnoula & McMurray, 2016).”

2) A potentially interesting analysis would be to extract a measure of lexical integration (i.e., Non-hermit RT - Hermit RT) for each child on Day 1 and Day 2 and test whether Day 1 score predicts Day 2 score. This type of analysis would allow one to see whether immediate/early integration is a predictor of post-sleep integration and whether this relationship is modulated by Delay and/or Vocabulary.

As above, we are not keen on running additional unplanned analyses. The model that we have presented already tells us that lexical competition on Day 2 (but not Day 1) is influenced by vocabulary *and* delay. Finding out whether lexical competition on day 1 predicts lexical competition on day 2 would be addressing a different question. Since readers will have access to the full raw dataset, we think it more appropriate to let others carry out their own additional exploratory analyses on the data. Furthermore, correlating two difference scores (non-hermit RT - hermit RT) is potentially problematic, given difference scores have a tendency to be unreliable and are heavily confounded by baseline RT (which changes from Day 1 to Day 2).

Minor suggestions:

1. On p. 15 (lines 14-15), it is reported that “RTs were faster overall on Day 2 (M = 1055 ms, SD = 179) than Day 1 (M = 1280 ms, SD = 192)”. It would be helpful to include the stats here, given that the difference was significant.

The stats for this have now been included in the manuscript (see p. 15).

2. On p. 15, the authors mention that “For the Long Delay group, there was a general trend of faster responses for participants with higher vocabulary scores [...]”. Again, this might be a good point to include the statistical output for the 2-way Delay × Vocabulary interaction.

We thank the reviewer for highlighting this. We have amended the text to clarify that this general trend was descriptive; there was no statistical difference between the slopes (see p. 15).

Recommendation:

Despite this being a very interesting and well-conducted study, I am afraid I cannot recommend it for publication in its current form. A number of concerns and suggestions are mentioned above, among which I can point to two major issues: 1) Given that one of the effects does not survive correction for multiple comparisons, it would be appropriate to tone down the conclusions based on this result and instead focus on the significant results. 2) I strongly suggest following-up on the 4-way (Vocabulary × Delay × Day × Word type) interaction in a different way, so that the results can be better linked to the hypotheses.

Reviewer #2

This is a really interesting study, investigating the relation between learning new words and sleep in 8 to 12 year old children, when the time interval between a study phase and sleep (during which consolidation of learning can take place) varies. The study tests an understudied group investigating effects of learning and memory that have potentially important practical implications for pedagogy, as well as theoretical models of sleep and memory. The

results failed to replicate previous studies on the consolidation effects of sleep, indicating no overall difference on lexical integration between immediate versus delayed sleep. However, explorative analyses demonstrated a significant interaction between immediate versus delayed sleep and vocabulary size. Children with smaller vocabularies benefited more from immediate sleep than delayed sleep in lexical integration. For children with larger vocabularies, there was no evidence difference between immediate and delayed sleep.

The lack of replication of previous studies on consolidation effects associated with sleep are effectively discussed in the paper, and the writing and presentation of the work is exemplary in its detail, reflective quality, and integration with the literature. The work is a useful addition to the literature, though the significant results in the paper represent small effects in complex interactions, and may be difficult to replicate.

The authors mention on p.9 that the four-way interactions between vocabulary size, delay, day, and item type is an explorative hypothesis, and the interaction just reaches a conventional level of significance. This is ok – that’s what conventions are for – but there are also a lot of filtering of outliers performed before the analyses are performed (pp.13-14). I would like the authors to also mention whether the results are robust to inclusion of outliers, so that the robustness of the four-way interaction can be ascertained. It is of course justified to remove outliers, but if the results change when some of these outliers are included, then that is useful additional knowledge, and does not affect the main message of the paper.

We appreciate the complex nature of the 4-way interaction and the reviewer is right to be concerned over how robust and replicable it is. However, we do not agree that re-running the model again with outliers included is an appropriate way to address this concern. Although we did not pre-register this particular study, we followed our lab protocol for outlier removal (as described in this pre-registration of the adult version of this study <https://royalsocietypublishing.org/doi/10.1098/rsos.181842>). The outliers are removed for good reason, and given the sample size, retaining outliers would more than likely influence the model in some way. We would then be left in a very difficult situation as to which model to follow. Adding in further models (with further problems of correcting for multiple comparisons) would only weaken the paper, in our opinion. Thus, we do not think this would be an example of good practice. As stated in the Discussion, this interaction certainly requires further investigation and replication, and we hope that others are encouraged to examine it. Incidentally, we are about to submit another paper on word learning in poor comprehenders which uses different training materials and comes to the same conclusion that learning closer to bedtime is beneficial for long-term memory of new words, but again only for children with weak vocabulary. Thus, whilst the 4 way interaction is weak, we have confidence that it is real and is deserving of further research, given the interesting theoretical and practical implications that arise from it.

Relatedly, can the thresholds for identifying outliers be justified? On p.13 participants with z-scores larger in magnitude than $-/+3.29$ are removed. Why this value? On p.14 participants with z-scores less than -2.5 were removed. Why this value? On p.14 participants with BPVS

z-scores greater than 2.5 were removed. Also it looks like scores < -2.5 were removed. Why this value?

We thank the reviewer for drawing our attention to this. We do state on p. 14 that the 2.5 outlier thresholds for accuracy rates, BPVS and within subjects RT trials were selected to be comparable to our previous published research (and many other studies). When we were preparing our analysis plan for this study, we learned that other groups were beginning to use dfbetas as a way to remove influential cases that exert undue influence over the parameters in the model. We decided to use this approach here, given the variability in child data and the likelihood that influential cases could have been influencing the model in a sample of this size. To adhere to the practice of colleagues who were already using dfbetas, and to take a conservative approach to avoid removing too many influential cases (especially given the relative infancy of this practice), we decided to adopt the 3.29 threshold, and stick with the 2.5 threshold for outlier removal to retain comparability with our previous / current outlier protocol. We have added the following clarification to the manuscript regarding dfbetas, p14 “We adopted a 3.29 threshold for the removal of influential cases (as opposed to the standard practice of a 2.5 threshold for outlier removal, as described below) to take a conservative approach and avoid the risk of unnecessary data loss.”

Throughout, the authors should refer to explorative analyses for additional analyses that were not initially planned during the execution of the paper. For instance, hypothesis 3 is explorative. Is hypothesis 4 also explorative? It seems equally complex. When hypotheses are being discussed with respect to the results, if the explorative nature of the analysis can also be mentioned that again helps the reader to navigate the research.

Thank you for this. We agree and have made amendments on p. 7 and p. 15 to make the nature of the hypotheses clearer. To note, hypothesis 3 (Day x Word Type x Vocabulary) is not exploratory because it is based on our previous findings. It's hypothesis 4 that's the exploratory one. Hopefully this is clearer now.

Appendix B

Dear Professor Dunn,

Re: Growing up with interfering neighbours: The influence of time of learning and vocabulary knowledge on written word learning in children

Thank you for the feedback on the above paper. We have addressed the comments and hope that you feel the manuscript is now suitable for publication in *Royal Society Open Science*. Of course, we are very happy to respond to any further comments. The details of how we have addressed the specific issues raised are below.

Yours sincerely,

Dr Sarah Walker

Reviewer: 2

Comments to the Author(s)

This is an excellent revision of the first version of the paper.

The authors have addressed my primary concern about the explorative analyses being clearly highlighted.

My other concern was about the identification of outliers. The authors have now taken two approaches - one based on dfbetas, and the other on identifying zscores for participants. These are both now mentioned on pp. of the paper, but it is not clear to me the order in which these were adopted (or even whether the omission of outliers using the zscores is superfluous given the use of dfbetas). In a final adjustment, please can the authors clarify the relation between these outlier analyses (the order in which they are applied, and why they are both still necessary)? It seems that the dfbetas analysis removes participants, whereas the zscores approach removes individual observations from participants (that at least is what the description on p.13 seems to suggest). The authors suggest this is more conservative, so is the removal via zscores of individual participant's responses still necessary? Stating this practice explicitly will be helpful in future studies that aim to use similar models.

We thank the reviewer for highlighting this issue with our description of outlier and influential case removal. We have now re-worded our analysis plan on pp. 13 and 14 to clarify the order of outlier and influential case removal, and also clarified why both are necessary in accordance with the recommendations of Nieuwenhuijs, te Grotenhuis and Pelzer (2012).